# Cardenolide toxin diversity impacts monarch butterfly growth and sequestration

**Anurag A Agrawal[1,2]\*, Amy P Hastings[1], Paola Rubiano-Buitrago[1]**

[1]Department of Ecology and Evolutionary Biology, Cornell University, Ithaca, United States; [2]Department of Entomology, Cornell University, Ithaca, United States

## eLife Assessment

This **important** study investigates how structurally diverse cardenolide toxins in tropical milkweed, especially mixtures containing nitrogen- and sulfur-containing variants, influence monarch caterpillar feeding, growth, and toxin sequestration. The experiments provide **solid** evidence that chemical diversity within a single group of plant toxins can have combined effects on even highly specialized herbivores that differ from the effects of each toxin alone. However, as the mixture design does not fully separate true diversity effects from the influence of the N,S-cardenolides themselves and the ecological basis for the chosen natural ratios remains weakly justified. As a result, the broader conclusions would require more fully justified concentration regimes, mixture treatments that exclude N,S-cardenolides, and tests on living plants and non-adapted herbivores to firmly support the proposed coevolutionary interpretation.

**Abstract** In coevolutionary interactions, host plants accrue novel chemical defenses that specialist herbivores counter by detoxification and sometimes sequestration. We recently found unusual nitrogen- and sulfur-containing (N,S-) cardenolides in some milkweeds—highly toxic compounds that monarch butterflies (*Danaus plexippus*) detoxify during sequestration. We hypothesized that the N,S-cardenolides in *Asclepias curassavica* (uscharin and voruscharin) would reduce caterpillar performance and sequestration more than other abundant related cardenolides (15-hydroxy-calotropin, frugoside, calactin). Cardenolides generally increased feeding relative to controls, but voruscharin was not stimulatory and substantially reduced growth efficiency. Exposure to either N,S-cardenolide produced the lowest sequestration and reduced sequestration efficiency, consistent with detoxification limiting toxin retention. We next tested whether toxin mixtures impose additional costs relative to individual compounds. We prepared two mixtures, one with equal concentrations of five cardenolides and a 'realistic mixture' reflecting natural proportions. Relative to the average of single compounds, mixtures reduced feeding, growth, sequestration, and sequestration efficiency, indicating phytochemical diversity effects exceeded expectations from an additive model. The two mixtures similarly reduced growth, but feeding on the realistic mixture yielded the lowest sequestration. We conclude that coevolution can produce highly specialized defense metabolites such as N,S-cardenolides that thwart even sequestering herbivores, and that phytochemical mixtures strengthen plant defense.

## Introduction

Individual plants produce thousands of the so-called secondary compounds, those with no known function in primary metabolism (i.e. resource acquisition and allocation), many of which are defensive

**\*For correspondence:**
aa337@cornell.edu

**Competing interest:** The authors declare that no competing interests exist.

**eLife digest** Monarch butterflies are often considered a pinnacle of adaptation, feeding exclusively on milkweed plants and having adapted to the milkweed's poisonous defenses. This relationship is a textbook example of an evolutionary arms race: milkweeds produce potent toxins called cardenolides, and monarchs have evolved not only to tolerate these poisons but also to accumulate them in their bodies. This accumulation, known as sequestration, protects monarchs from predators.

While cardenolides represent a classic case of a defense overcome by a specialist herbivore, ongoing coevolution suggests that plants may continue to evolve increasingly potent strategies. Indeed, plants rarely produce just one toxin, and previous research has shown that some milkweeds produce highly toxic nitrogen- and sulfur-containing cardenolides.

Agrawal et al. investigated whether these chemicals impose greater costs on monarch caterpillars compared to other common cardenolides. The researchers isolated and purified five dominant cardenolide toxins from the tropical milkweed, *Asclepias curassavica.* To test whether consuming a realistic mixture of toxins impairs caterpillar growth and defense sequestration more than consuming individual toxins in isolation, the researchers fed different combinations of the compounds to caterpillars.

This revealed that in isolation, nitrogen- and sulfur-containing cardenolides were not stored in their original molecular composition. Instead, caterpillars metabolized them into less toxic forms, reducing their ability to accumulate defensive chemicals compared to when fed other toxins (without nitrogen and sulfur). Caterpillars fed a mixture of all five toxins grew more slowly and sequestered fewer toxins than those fed equal amounts of single compounds. These results suggest that chemical diversity itself is a powerful plant defense, likely by overwhelming the detoxification systems of specialist herbivores.

The study of Agrawal et al. provides key insights into how plants defend themselves against pests that have adapted to their toxins. These findings could inform the development of more effective, multi-component pest management strategies in agriculture, mimicking nature's "cocktail approach" rather than relying on single chemicals. Future research should extend beyond laboratory assays to test these effects on living plants and across a diversity of insect herbivores to validate these ecological theories further.

(*Fraenkel, 1959*). Although the benefits of plant defensive chemistry are well established (*Mauricio and Rausher, 1997*; *Schoonhoven and Dicke, 2005*; *Agrawal, 2011*), why plants produce such a diversity of secondary compounds has long been a mystery (*Romeo et al., 1996*; *Rasmann and Agrawal, 2009*; *Speed et al., 2012*). Defenses from distinct chemical classes (e.g. protease inhibitors and alkaloids) can interact in myriad ways (*Steppuhn and Baldwin, 2007*), but even within a class, a remarkable amount of functional and structural diversity exists. When diversity exists within a class of compounds sharing the same mode of action (e.g. cyanogenic glycosides, cardenolides, or ellagitannins) and when deployed in the same plant tissues, a defensive benefit of deploying multiple compounds is often predicted; nonetheless, the proximate and ultimate explanation for this diversity is not well understood (*Richards et al., 2016*; *Kessler and Kalske, 2018*; *Wetzel and Whitehead, 2020*). Are individual compounds targeting different plant attackers, or do mixtures act as a more effective defense than individual compounds alone?

Within a defense class, variation in the biological activity and ultimately the potency of a particular compound against herbivores may be derived from structural attributes (*Berenbaum and Zangerl, 1996*; *Romeo et al., 1996*; *Macel et al., 2005*; *Zaman et al., 2025*). For example, alkaloids are perhaps the best-studied group of phytochemicals, with known biosynthetic pathways, structural modifications (e.g. glycosylation, oxidation, annulation), and differential impacts on biological targets (*Bhambhani et al., 2021*). Such structural variation impacts the molecular complexity, physicochemical properties (e.g. polarity), and potentially specific interactions with the physiological target in an animal consumer. As a case in point, among cardenolide toxins produced by milkweeds (*Asclepias* spp., Apocynaceae), all having the same mode of action (binding to Na/K-ATPase in animal cells) (*Agrawal et al., 2012b*), hundreds of compounds have been identified, ranging from 350 to 1066 Da and with a broad range of polarity and structural features (*Rubiano-Buitrago et al., 2026*). In particular, only about 5% of milkweed cardenolides have major structural modifications beyond the steroidal core, unsaturated lactone, and sugar moieties, and these have been proposed as the most complex and

**Figure 1.** A proposed biosynthesis pathway for cardenolides of *A. curassavica* built on coroglaucigenin (*Rubiano-Buitrago et al., 2026*). Modifications are indicated by purple highlighting, and each arrow indicates a hypothesized step; multiple arrows indicate multiple concerted reactions without displaying all the intermediates. All compounds, except the genin, are known to occur in *A. curassavica*; however, gofruside is rarely found in high quantities in the foliage (*Roy et al., 2005*; *Rubiano-Buitrago et al., 2022*).

The online version of this article includes the following figure supplement(s) for figure 1:

**Figure supplement 1.** Inhibition curves estimated from six concentrations of 15-hydroxy-calotropin on a sensitive (porcine) and adapted (monarch) sodium-potassium ATPase in vitro.

potent cardenolides (*Rubiano-Buitrago et al., 2026*). Our previous work has shown >1000-fold variation in the in vitro functional toxicity of different cardenolides on different herbivore target enzymes (*Agrawal et al., 2021*; *Agrawal et al., 2022*). Nonetheless, we are still looking for a predictive framework for understanding the diversity of cardenolide compounds and their ecological impacts.

In addition to individual toxins varying in their biological activity, we still understand relatively little about their combined effects when produced as mixtures (as they are typically consumed in nature). When presented together, secondary compounds may sum to a greater impact on herbivores than can be predicted by any single effect (termed a *phytochemical diversity effect*) (*López-Goldar et al., 2024*). Experimental approaches to studying effects of phytochemical diversity within a compound class have been increasing and are often conducted from the herbivore's perspective (*Berenbaum and Zangerl, 1996*; *Richards et al., 2016*; *Whitehead et al., 2021*; *López-Goldar et al., 2024*; *Zaman et al., 2025*). We still know rather little about the impact evenness within mixtures, and whether small quantities of particular compounds may shape the net biological effect within mixtures (*Glassmire et al., 2020*; *Wetzel and Whitehead, 2020*; *López-Goldar et al., 2024*). In summary, despite the widespread occurrence of substantial phytochemical diversity within a chemical class and even in a single plant species, substantial work remains to enhance the methods, realism, and predictability such that the general consequences of phytochemical diversity can be evaluated.

Here, we take an organismal approach to test effects of phytochemical diversity on the monarch butterfly (*Danaus plexippus*), with a special focus on structurally diverse cardenolide toxins. In particular, we address key challenges in advancing our understanding of phytochemical diversity by (1) isolating the specific compounds from a single plant species, (2) administering the compounds in a realistic way to an herbivore adapted to the host plant, and (3) controlling total toxin concentrations across treatments to allow for direct comparisons without confounding toxin concentration. We have been studying the impacts of specific toxins using isolated and purified cardenolides from the foliage of tropical milkweed (*Asclepias curassavica*), a major host plant of the monarch (*Figure 1*). In particular, unusual compounds with nitrogen- and sulfur-containing heterocycles (e.g. uscharin and voruscharin, hereafter N,S-cardenolides) are highly complex, non-polar, and the most toxic compounds known on the monarch Na/K-ATPase (*Agrawal et al., 2021*; *Agrawal et al., 2024b*). Nonetheless, the in vivo impacts of these compounds on growth, sequestration, and physiological efficiencies (e.g. mass gained, or toxins stored per unit leaf consumed) have not been previously studied.

Here, we start by addressing the relative impact of five dominant cardenolides from *A. curassavica* on monarch growth and sequestration individually. Accordingly, we first test the hypothesis that these

**Table 1.** Structural and functional features of the isolated cardenolides used in this study, comprising the dominant compounds in *A. curassavica* (70% of the total leaf cardenolides).

The structural complexity values are based on dox-g (*Rubiano-Buitrago et al., 2026*). We also provide a non-chromatography-dependent metric of polarity, WLOGP (*Daina et al., 2017*). IC-50 represents µM needed to inhibit the monarch Na/K-ATPase by 50% in vitro, data from *Agrawal et al., 2021*, except for 15-hydroxy-calotropin which was generated for this study (*Figure 1—figure supplement 1*, *Supplementary file 1*). The proportion of total cardenolides in *A. curassavica* leaves is based on past work (see Materials and methods). Proportion in 'real mix' was scaled up from the proportion found in leaves to sum to 1.

|  | 15-Hydroxy-calotropin | Frugoside | Calactin | Uscharin | Voruscharin |
|---|---|---|---|---|---|
| Nitrogen-cardenolide | No | No | No | Yes | Yes |
| Molecular mass | 548.6 | 536.7 | 532.6 | 587.7 | 589.7 |
| Structural complexity (dox-g) | 20 | 17 | 19 | 22 | 21 |
| HPLC-DAD retention time (min) | 11.65 | 12.66 | 15.93 | 20.59 | 21.25 |
| WLOGP | 0.97 | 1.43 | 2 | 2.77 | 2.29 |
| IC-50 on monarch enzyme (µM) | 64.8 | 27.8 | 3.2 | 1.0 | 2.0 |
| Prop. total cards in leaves | 0.10 | 0.05 | 0.10 | 0.10 | 0.35 |
| Prop. total cards in 'real mix' | 0.14 | 0.07 | 0.14 | 0.14 | 0.50 |

N,S-cardenolides (uscharin and voruscharin, comprising up to 50% of total foliar cardenolides) reduce monarch feeding, growth, and digestive efficiency compared to three other cardenolides from *A. curassavica*. We expected these compounds to be broken down to less toxic forms before sequestration (*Agrawal et al., 2021*; *Agrawal et al., 2024b*), potentially reducing sequestration overall and sequestration efficiency compared to other cardenolides. Finally, we tested the phytochemical diversity hypothesis, predicting that cardenolide mixtures would reduce growth and sequestration more so than equimolar concentrations of single compounds. We address this diversity hypothesis with two types of mixtures: equal concentrations of the five compounds vs. a more realistic mixture with the compounds in proportion to their relative abundance in *A. curassavica* leaves. In sum, this work addresses the impact of a natural diversity of milkweed toxins on a native and highly adapted herbivore via impacts on digestive and defensive ecophysiology.

## Results

### Characterizing structural diversity of isolated cardenolides

The five dominant cardenolides of *A. curassavica* are relatively similar in molecular weight (≈10% difference between lowest and highest), but vary more substantially in our estimates of their chemical complexity (29%), polarity (WLOGP: 64%, retention time: 82%), and toxicity to the monarch enzyme (28-fold) (*Table 1*). In particular, frugoside and calotropin are part of the proposed biochemical pathway leading to the production of the N,S-cardenolides voruscharin and uscharin, while 15-hydroxy-calotropin is a small structural modification of calotropin (with a hydroxyl group added, *Figure 1*). Among the group, the N,S-cardenolides are the heaviest, most non-polar, most complex, and have the highest in vitro inhibitory capacity of the monarch Na/K-ATPase compared to the other three compounds (*Table 1*).

### Effects of single cardenolides: growth and feeding

Compared to monarchs feeding on control leaves, experimental addition of individual cardenolides increased caterpillar growth rates, and this effect was significant for frugoside, calotropin, and N,S-containing uscharin, but not the other N,S-containing cardenolide, voruscharin, or for 15-hydroxy-calotropin ($F_{5,91}=3.985$, $p=0.003$, *Figure 2A*). Across all treatments, individual caterpillars grew more when they ate more leaf tissue, and the slopes of these relationships were the same across treatments (predicting growth: dry mass consumed $F_{1,80}=245.87$, $p<0.001$; cardenolide treatment $F_{5,80}=1.04$, $p=0.399$; interaction $F_{1,80}=0.465$, $p<0.802$, *Figure 2—figure supplement 1A*). In other words, three of the five compounds stimulated feeding, including the most potent N,S-cardenolide.

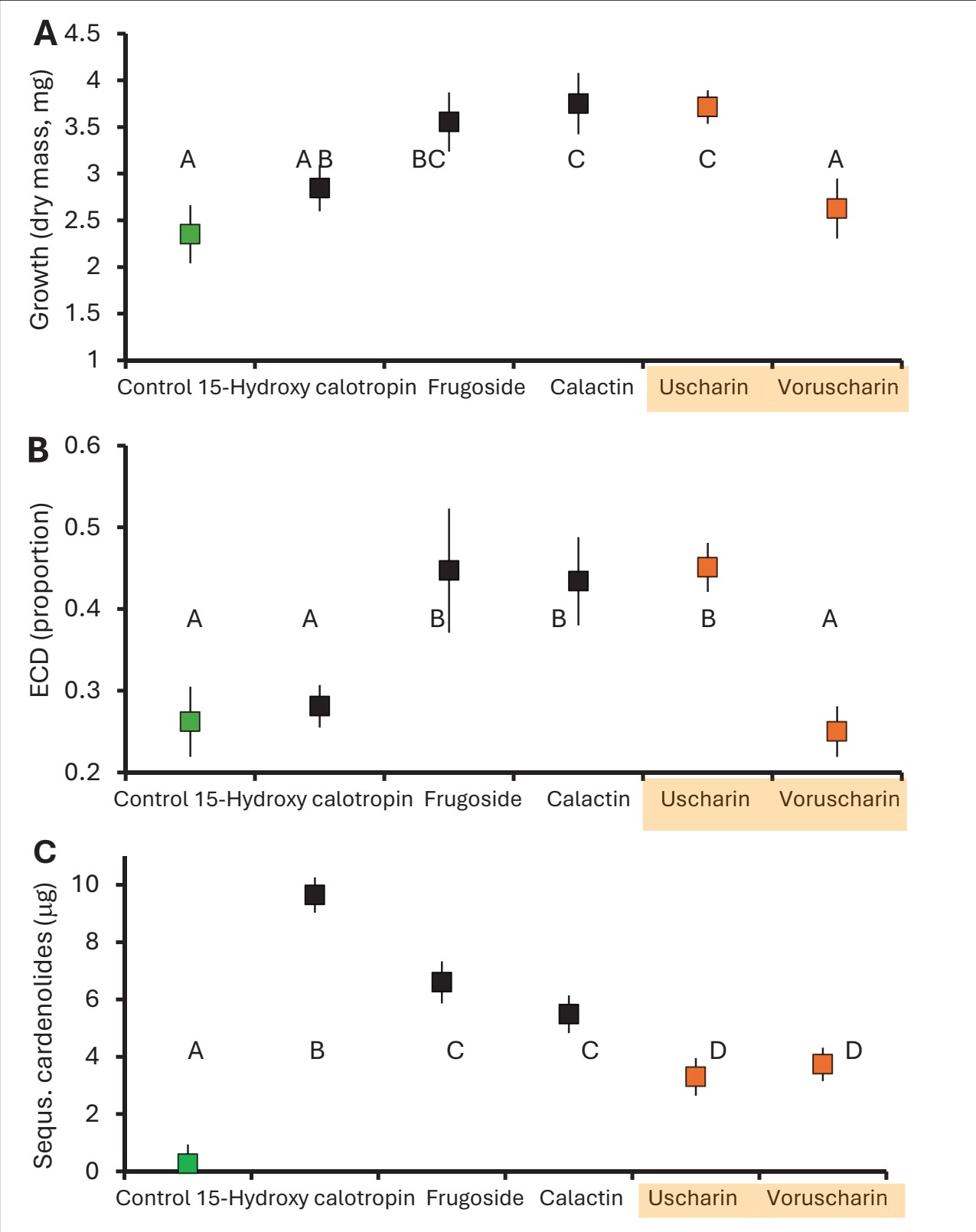

**Figure 2.** Monarch caterpillars are impacted by feeding on *Asclepias incarnata* leaves painted with isolated cardenolides from *A. curassavica*. (**A**) Growth after 9 days (most caterpillars in the third instar), (**B**) efficiency of conversion of digested matter, and (**C**) total cardenolides sequestered. Shown are means ± SEs and different letters indicate a significant difference (p<0.05, Fisher's LSD). The green symbol has no cardenolides added (*A. incarnata*), and the two orange symbols represent cardenolides with N,S-ring moiety.

*Figure 2 continued on next page*

*Figure 2 continued*

The online version of this article includes the following figure supplement(s) for figure 2:

**Figure supplement 1.** Monarch caterpillars are impacted by feeding on leaf discs painted with isolated cardenolides from *A. curassavica*.

The amount of frass excreted, a component of digestion, was differentially impacted by our cardenolide treatments; in particular, when consuming N,S-containing uscharin, less frass accumulated per leaf mass consumed than for the four other cardenolides (leaf mass consumed $F_{1,79}$=180.48, p<0.001; treatment $F_{5,79}$=2.48, p=0.036; interaction $F_{1,79}$=2.43, p=0.042, *Figure 2—figure supplement 1B*). Nonetheless, the efficiency of conversion of digested matter (ECD, the proportion of assimilated food converted to biomass; caterpillar growth/[ingested leaves – frass]) was not impacted by uscharin; rather, the cardenolides that stimulated feeding and growth also increased this conversion efficiency (up to >40%, $F_{1,85}$=5.136, p<0.001, *Figure 2B*), while monarch caterpillars feeding on voruscharin and 15-hydroxy-calotropin (which showed lower growth) maintained low ECDs (≈25%), equivalent to that of controls (*Figure 2B*).

## Effects of single cardenolides: sequestration

Our treatments with applied cardenolides to *A. incarnata* well approximated the amount of cardenolides in leaves and sequestered by monarchs when feeding on *A. curassavica* (*Figure 3—figure*

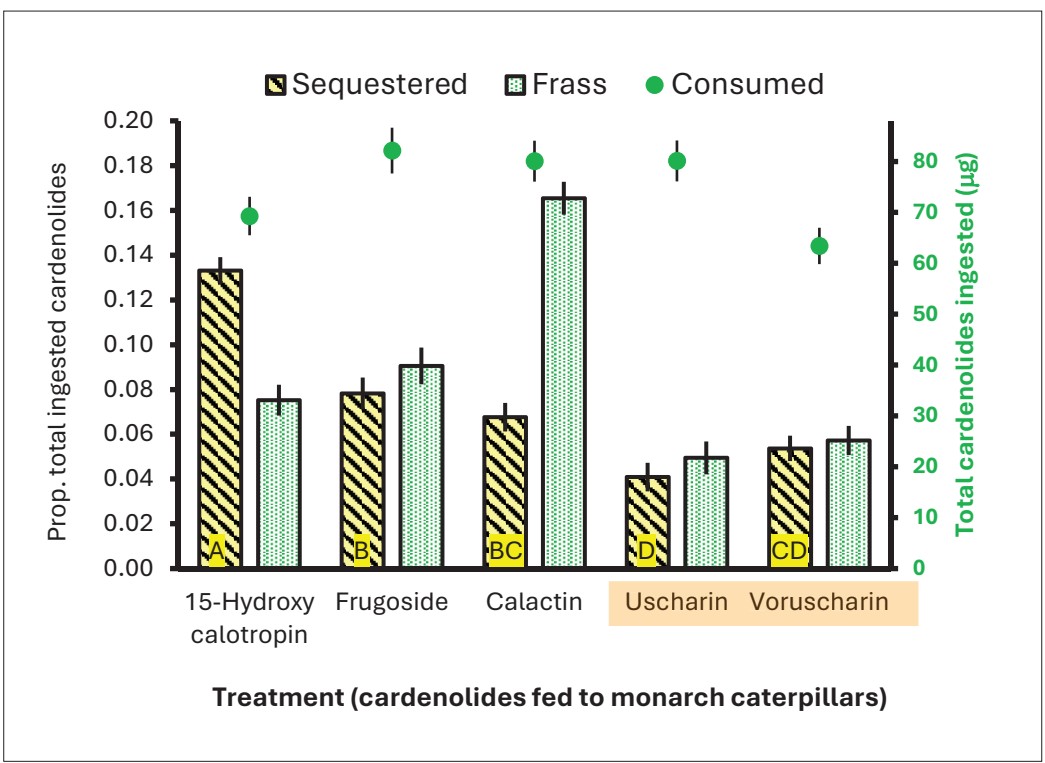

**Figure 3.** Monarch caterpillars differentially sequester and excrete cardenolides when eating leaves painted with isolated compounds from *A. curassavica*. Shown are means ± SEs of the proportion of total cardenolides ingested that are sequestered or excreted (bars, left axis); shown also is the amount of cardenolide ingested (green dots, right axis). 15-Hydroxy-calotropin, frugoside, and calactin are sequestered intact, while uscharin and voruscharin are stored after conversion to calotropin and calactin. Orange shading indicates cardenolides with N,S-ring moiety. Data on cardenolide concentrations sequestered and excreted on a mass basis are given in *Figure 2*. Different letters indicate a significant difference between treatments for sequestration efficiency (p<0.05, Fisher's LSD).

The online version of this article includes the following figure supplement(s) for figure 3:

**Figure supplement 1.** Monarch caterpillars differentially sequester ($F_{6,102}$=102.22, p<0.001) and excrete ($F_{6,100}$=37.12, p<0.001) cardenolides when feeding on leaf discs painted with isolated compounds from *A. curassavica*: Shown are means ± SEs of cardenolide concentration (on a per body mass basis), and different letters indicate a significant difference (p<0.05, Fisher's LSD, letters are only comparable within a tissue type).

*supplement 1*). For each of the three cardenolides lacking a nitrogen moiety (15-hydroxy-calotropin, frugoside, calactin), monarchs sequestered the compound intact (>90% of the stored cardenolides were those applied to the leaves); conversely, uscharin and voruscharin were both converted to calotropin (>56% of the stored cardenolides) and calactin (>20% of the stored cardenolides). Neither uscharin nor voruscharin was detected in caterpillar bodies, and only uscharin was detected in frass (comprising 20% of excreted cardenolides for caterpillars fed uscharin).

When feeding on the five compounds individually, monarch caterpillars sequestered the least total toxins when eating the two N,S-cardenolides, which were converted to other compounds (*Figure 2C*). Overall, caterpillars ingested between 63 and 83 µg of cardenolide, of which we were able to account for ≈15% in the body and frass (*Figure 3*); unaccounted cardenolides may be degraded, which has been previously suggested (*Seiber et al., 1980*). Critically, when feeding on uscharin or voruscharin, caterpillars had the lowest sequestration efficiency and excreted the least cardenolides compared to other treatments (contrast of non-N,S vs. N,S-cardenolides for proportion sequestered $F_{1,73}$=64.490, p<0.001; proportion excreted $F_{1,73}$=76.00, p<0.001, *Figure 3*). Thus, the N,S-cardenolides are not sequestered intact, and after conversion to non-N,S-cardenolides, they are sequestered relatively poorly, as well as excreted, indicating that more of these compounds are missing and possibly degraded. Among the other cardenolides, substantial variation exists, with the monarchs sequestering twice as much 15-hydroxy-calotropin as they excreted, and the reverse for calactin, excreting

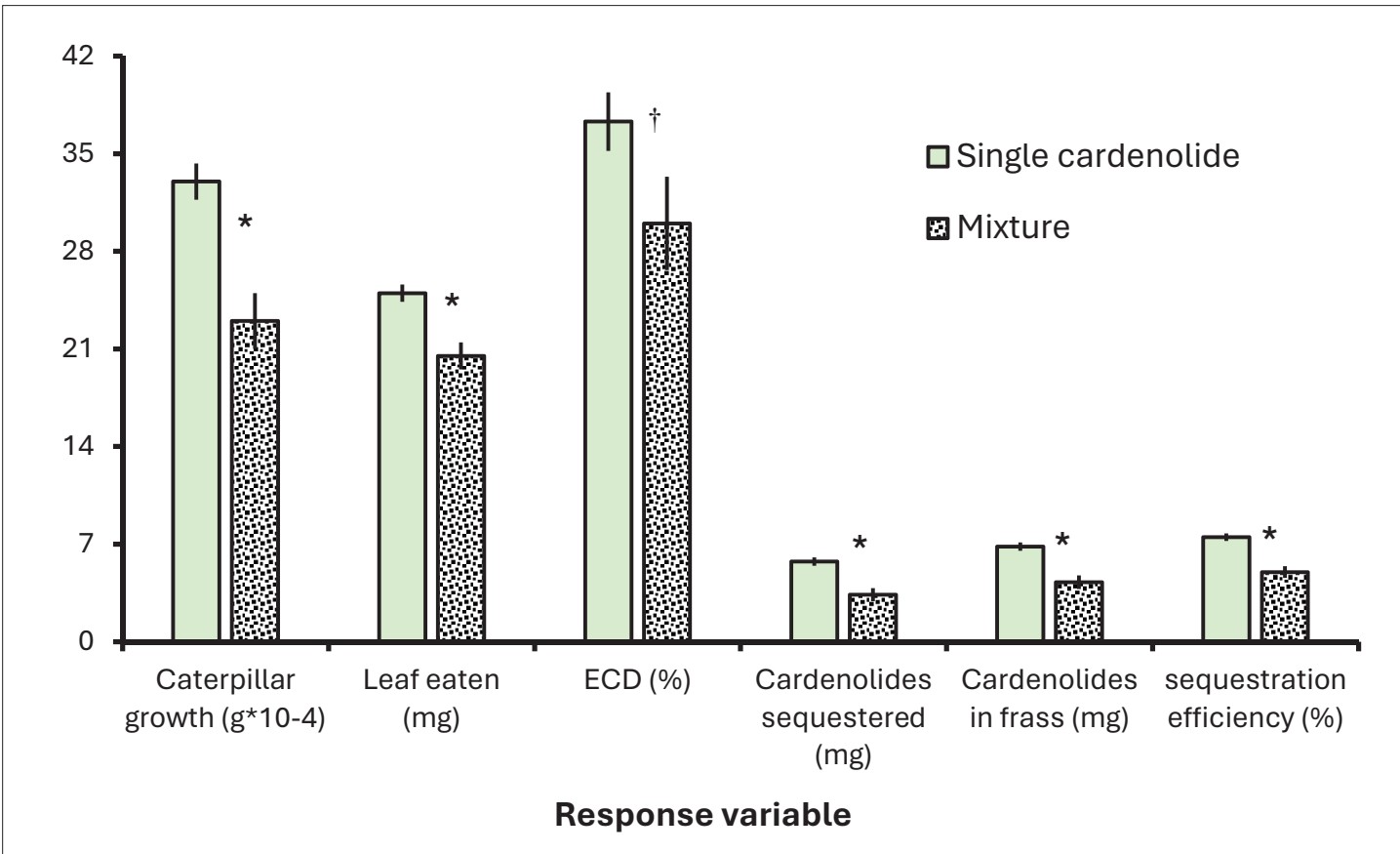

**Figure 4.** Monarch caterpillars differentially grow and sequester cardenolides when feeding on single isolated compounds from *A. curassavica* compared to mixtures. Shown are means ± SEs for several response variables with the units shown below the X axis. Sequestration efficiency was calculated by the mg of cardenolide ingested/mg cardenolide sequestered. Single effects are the average of the five compounds administered individually, whereas the mixture is the average of an equal mixture treatment and realistic mixture treatment. In all cases, total cardenolide concentrations were presented to caterpillars on an equimolar basis. Significance (p<0.01) is shown by *, while † indicates p=0.07. All treatment means are provided in *Figure 3—figure supplement 1*.

The online version of this article includes the following figure supplement(s) for figure 4:

**Figure supplement 1.** Mean ± SE of nine treatments growing monarch caterpillars on two controls (green: positive on *A. curassavica* and negative on *A. incarnata*), three non-N,S-cardenolides (gray), two N,S-cardenolides (orange), and two types of mixtures (striped).

twice as much as they sequestered (*Figure 3*). In summary, sequestration was consistently worse for the N,S-compounds, likely because detoxification is required; however, the strongly disparate effects on feeding and growth were not predicted by the presence of the N,S-moiety.

## Effects of mixtures

The effects of cardenolide mixtures were assessed first by contrasting the average of all single cardenolide treatments against the two mixtures for each response variable. In short, all feeding, growth, and sequestration parameters of monarch caterpillars were reduced by feeding on cardenolide mixtures (*Figure 4*). For growth, the negative effect of mixtures vs. single compounds was driven primarily by reduced feeding (i.e. deterrence); in other words, when we include the amount dry mass tissue consumed in the analysis, the effect of mixtures is no longer significant in *Figure 4* (p=0.173).

Perhaps most interestingly, the efficiency of sequestration of ingested cardenolides declined from 7.5% singly to 5% in mixtures. This result controls for feeding, as we used the estimated micrograms of cardenolides ingested to calculate the efficiency. For this metric, we constructed a specific contrast for the expected values based on the proportions used in the equimolar and in the realistic mixtures (*Table 1*). The sequestration efficiency of the equimolar mixture was reduced by 18% compared to the average of the single compounds (contrast: p=0.036) and was reduced by 42% in the realistic mixture compared to the expected values based on relative abundances of the five compounds (p=0.002).

Overall, even if not statistically significant, we note that monarch caterpillars raised on the realistic *A. curassavica* cardenolide mixture had the lowest growth and feeding, and the lowest cardenolide sequestration, excretion, and sequestration efficiency compared to all single treatments, supporting a hypothesis of adaptive deployment of cardenolide mixtures by *A. curassavica* (*Figure 4—figure supplement 1*).

## Discussion
### Effects of individual compounds

As is the case for most plant species, milkweeds contain complex mixtures of secondary metabolites implicated in defense against herbivores. With the advent of improved chromatographic techniques (*Pandohee et al., 2023*), more is being discovered about such mixtures, especially on the diversity of compounds within chemical classes. With >50 cardenolide structures reported from *A. curassavica* (*Roy et al., 2005*; *Li et al., 2009*; *Zhang et al., 2014*; *Rubiano-Buitrago et al., 2022*), we have been investigating the diversity of these compounds and their effects. These toxins are spread across plant organs, and many compounds occur in very small amounts; here, we have focused on five dominant cardenolides from foliar tissues. Despite the fact that N,S-cardenolides have been known for decades, their importance in toxicity was discovered only recently (*Hesse et al., 1950*; *Reichstein et al., 1968*; *Seiber et al., 1978*; *Benson et al., 1979*; *Züst et al., 2019*). Five years ago, we reported that uscharin and voruscharin were not sequestered intact, but converted to less toxic calotropin and calactin, and imposed a cost of sequestration for monarch caterpillars (*Agrawal et al., 2021*). Perhaps most surprisingly, in in vitro assays, these same compounds had relatively weak effects on a sensitive Na/K-ATPase enzyme, suggesting that they are targeted at the adapted specialists' more resistant pump (*Rubiano-Buitrago et al., 2026*). These findings are reminiscent of classic work on coumarins, where linear furanocoumarins are tolerated by specialist black swallowtail caterpillars, but more unusual angular furanocoumarins, which are not toxic to many organisms, specifically negatively impact black swallowtails (*Berenbaum and Feeny, 1981*). Later, it was found that these two types of furanocoumarins synergize to impact the specialist (*Berenbaum and Zangerl, 1996*). Despite this well-characterized example, we still know relatively little about the structural diversity of chemical defenses within a compound class and their roles against most specialist herbivores (*Marquis and Koptur, 2022*).

In the current study, we confirm that N,S-cardenolides are not sequestered by monarchs and move to demonstrate that the processing of these compounds is associated with reduced sequestration efficiency and reduced cardenolide excretion compared to closely related non-N,S-compounds. One previous study fed isolated cardenolides to monarchs and reported the effects on sequestration and excretion (*Seiber et al., 1980*). In both studies, a substantial fraction (≈80%) of cardenolides was unaccounted for, with monarchs sequestering on the order of 10% of the cardenolides they consume, smaller amounts excreted, but a large portion potentially degraded by the caterpillar. Thus, although

many cardenolides are sequestered intact by monarch caterpillars, it is clear that some are detoxified, as in the case of N,S-compounds, where the removal of the N,S-ring substantially reduces toxicity (*Agrawal et al., 2021*; *Agrawal et al., 2022*). Likewise, several di-glycosides are stored as mono-glycosides by both monarchs and the seed-feeding milkweed bug *Oncopeltus fasciatus* (*Agrawal et al., 2022*; *Rubiano-Buitrago et al., 2023*). Finally, *Seiber et al., 1980*, reported that genins (cardenolides without any sugar moiety) were not sequestered intact, but were rather stored as more polar metabolites, as was the glycoside uscharidin, neither of which were part of the current study.

In our current work, the addition of cardenolides to a diet nearly devoid of the compounds (leaf discs of *A. incarnata*) stimulated caterpillar feeding and growth. Although at first glance this result can seem counterintuitive, plant secondary metabolites, even when effective defenses, often function as stimulants for feeding and oviposition by specialist insects (*Renwick, 2002*; *Schoonhoven and Van Loon, 2002*; *Bernays and Singer, 2005*; *Wink, 2018*). Although the extent of feeding stimulation was not predicted by whether the cardenolides were N,S-containing compounds, we found that the most stimulatory compounds not only increased caterpillar growth, but also increased the efficiency of conversion of ingested matter (ECD). Work thus far on monarch-cardenolide interactions has employed isolated compounds that were added to diets to study their effects; an alternative approach might be to use genetic transformation to administer different compounds in plants to insects. Although each approach has limitations, and often genetic modification has off-target effects on secondary metabolism, we acknowledge that our delivery of compounds was not on intact plants.

## Sequestration efficiency

Sequestration efficiency is the proportion of ingested plant toxins that are stored by herbivores. In our study, N,S-cardenolides had the lowest sequestration efficiency (*Figure 4*), presumably due to the detoxification required. Additionally, we found twofold variation among the three related non-N,S compounds (*Figures 1 and 4*). Remarkably, sequestration efficiency on our positive control, *A. curassavica* leaves, was the same as the average of our individual compounds (*Figure 3—figure supplement 1*). And finally, monarch caterpillars had 100-fold higher sequestration efficiency on our negative control (*A. incarnata*). This result is consistent with evidence demonstrating the upregulation of transporters and cardenolide concentration on this extremely low cardenolide species (*Jones and Agrawal, 2019*; *Tan et al., 2019*).

Sequestration efficiency may be determined by food intake rate, the extent of toxin processing (detoxification or transport) needed, and existing levels of the toxin already sequestered (*Jones et al., 2019*; *Beran and Petschenka, 2022*). For example, fertilization of tropical milkweed resulted in monarch caterpillars growing more slowly and having lower cardenolide sequestration efficiency than when feeding on controls (*Tao and Hunter, 2015*). This effect seems to have been driven by an overall negative relationship between foliar cardenolide concentrations and sequestration efficiency. In this experimental approach, across other systems (e.g. *Bowers and Collinge, 1992*; *Lampert, 2020*), and ours, caterpillar feeding and growth rates are often correlated with sequestration efficiency. These relationships may be due to general vigor, consumption rate, and exposure to the plant toxins or detoxification (and its associated costs). Thus far, in no plant-herbivore system have these relationships been disentangled.

## Phytochemical diversity effects

Mechanistically, defensive compounds may act at different stages of attack by herbivores and may reach different tissues in the consumer, ultimately providing greater defense than any one compound. For example, defensive phytochemicals can differentially impact herbivore oviposition, larval feeding, growth, sequestration, and mating (*Landolt and Phillips, 1997*; *Macel and Vrieling, 2003*; *Macel et al., 2005*; *Kim and Jander, 2007*; *Müller et al., 2010*; *Agrawal et al., 2021*). Additionally, the interaction diversity hypothesis posits that phytochemical diversity may not necessarily have non-additive effects on any one herbivore, but different compounds may target different species of attackers, thereby creating a benefit of phytochemical diversity against diverse insect pests (*Whitehead et al., 2021*; *Zaman et al., 2025*). Early on, *Berenbaum et al., 1991*, not only speculated that phytochemical diversity may be defensively beneficial via different pathways, but also that structural diversity within a compound class, in their case furanocoumarins, may be driven by environmental variation; such condition-dependent effects may both be caused

by available nutrients (e.g. *Tao and Hunter, 2015*; *Agrawal et al., 2024a*), as well as distinct benefits of chemical diversity in different environments (*Berenbaum et al., 1991*; *Agrawal et al., 2012a*).

Other benefits of phytochemical diversity from the plant's perspective can be manifold. Although 'synergism' between phytochemicals is often invoked, there is little to no evidence of synergistic effects of plant defense compounds in a pharmacological sense, especially when they are in the same chemical class (see *Table 1* in *López-Goldar et al., 2024*). Because some researchers interpret 'synergism' with such a biochemical mechanism, we demur from using this term, as synergism per se has not been well studied. Nonetheless, diversity effects, here defined as greater than additive impacts of multiple compounds compared to singletons, have been reported in several studies within a phytochemical class. In classic work, angular furanocoumarins showed relatively low toxicity in isolation but had greater than additive effects with linear furanocoumarins, both in specialists and generalist herbivores (*Berenbaum and Zangerl, 1996*). For amides and imides from *Piper* spp., evidence suggests that diversity effects depend on the level of specialization of herbivores and the type of assay (growth, parasitism, final size) (reviewed in *Richards et al., 2016*). Some studies with *Jacobaea* pyrrolizidine alkaloids provide evidence for greater effects in mixture compared to single compounds, although this effect was primarily found with sub-organismal (in vitro) assays (*Nuringtyas et al., 2014*). Our work demonstrates non-additive effects of two mixture types on monarchs and thus contributes to this ongoing synthesis of phytochemical diversity effects.

Two studies have measured impacts of phytochemical diversity not only on caterpillar performance and sequestration, but also on immunity to parasites. *Richards et al., 2012*, found that specialist *Junonia* caterpillars grew faster, had lower mortality, sequestered higher concentrations, and had reduced immune responses on artificial diets containing mixtures of two of iridoid glycosides (acubin and catalpol) compared to those containing individual compounds. In other words, this study reported a lower-than-expected defensive function of mixtures, but this may come with reduced immunity of caterpillars to enemies. In our studies of *A. curassavica*, we find the reverse: not only do mixtures of cardenolides reduce caterpillar performance and sequestration, but monarchs have greater immunity against the protozoan parasite. More work will be needed to address the causes of these effects and trade-offs between performance, sequestration, and immunity (*Hoogshagen et al., 2024*).

In previous research (*López-Goldar et al., 2024*), we demonstrated effects of phytochemical diversity using three abundant cardenolides from common milkweed. These effects were most evident in terms of a dominance effect of highly inhibitive compounds at the target site (Na/K-ATPase), as well as physiological effects of phytochemical diversity in assays with transgenic flies that were tolerant of cardenolides but lacked other forms of specialization. When diverse cardenolides were briefly fed to monarch caterpillars, the main cost of chemical diversity was through an enhanced cost of sequestration compared to when monarchs were feeding on single compounds. In the current work, diversity of cardenolides imposed multiple limitations for caterpillar development, spanning growth and sequestration efficiency. The only work on non-adapted herbivores on milkweeds and their diverse cardenolides was that on wild-type *Drosophila*, which were impacted by mixtures compared to their expected effects (*López-Goldar et al., 2024*). Future work directly contrasting adapted and unadapted herbivores will likely shed light on the mechanisms by which different organisms cope with phytochemical diversity.

Empirical evidence from several studies (*Berenbaum et al., 1991*; *López-Goldar et al., 2024*; *Zaman et al., 2025*) suggests that relatively minor constituents of the natural blend of compounds in leaves may be contributing to the observed toxicity and greater effect of mixtures compared to single compounds. Even when the mixture is not more toxic than the most toxic individual compounds, mixtures are often more than additive in their effects. This distinction is not based on mechanism, and therefore we have not been using the phrase synergistic, but rather 'non-additive diversity effect' in a statistical sense. This distinction is parallel to the plant diversity literature in ecology which aims to separate multiple mechanisms of diversity effects (*Petchey, 2003*). Our key point is that non-additivity in phytochemical defense may be adaptive because small amounts of highly toxic compounds may complement, synergize, or dominate the mixture, thereby providing an economic advantage in terms of potentially lower production costs with the benefits of diversity (*López-Goldar et al., 2024*).

## Conclusion

In milkweeds, as in many other groups, diversification of secondary metabolites appears to occur via the chemical evolution of glycosylation, oxidation, cyclization (i.e. ring formation), and addition of complex functional groups onto core existing chemical structures (*Figure 1*). For the group of coroglaucigenin-based cardenolides, molecular mass, complexity, polarity, and toxicity progress through each biosynthetic step leading to N,S-cardenolides (*Rubiano-Buitrago et al., 2026*). Although we have only tested one herbivore species here, our past work has shown similar effectiveness of these compounds against seed-feeding specialists as well, especially in vitro (*Agrawal et al., 2021*; *Agrawal et al., 2022*). Here, two N,S-cardenolides had unequal effects, with only uscharin having a stimulatory effect on feeding and growth; nonetheless, these compounds were both modified and showed poor sequestration efficiency of the resulting cardenolides. When in a realistic mixture, monarch caterpillars fed and grew the least and sequestered the least compared to caterpillars feeding on equimolar concentrations of any of the five single compounds (*Figure 3—figure supplement 1*). Thus, cardenolide mixtures presented in *A. curassavica* appear to be a compromise between reducing growth and sequestration of highly specialized monarchs and providing benefits to the monarch in fighting off enemies (*Hoogshagen et al., 2024*).

# Materials and methods

As our main goal was to compare the major cardenolides isolated from *A. curassavica* alone and in mixture, we took the approach of painting equimolar concentrations of total cardenolides in all treatments (*Table 1*). These painting treatments were imposed on leaf discs of *A. incarnata*, a closely related species (both species within the Incarnatae clade), but with cardenolide abundance at trace levels (*Agrawal et al., 2015*). Fully randomized within our experiment were a negative and positive control fed to caterpillars, *A. incarnata* painted with ethanol alone, and *A. curassavica* leaves, respectively. Although these treatments were not included in most statistical analyses comparing the different individual cardenolides or comparing single compound and mixtures, we specifically compare these treatments to controls where appropriate (i.e. *A. incarnata* alone in the analysis of growth and *A. curassavica* in the analysis of sequestration).

## Plant growth

Seeds of *A. incarnata* (collected from Tompkins Co., NY, USA) and *A. curassavica* (purchased from Everwilde Farms, Fallbrook, CA, USA) were surface-sterilized with 10% bleach for 10 min, rinsed, nicked, and stratified at 4°C in Petri dishes lined with moist paper towels for 7 days. Seed dishes were then moved into an incubator at 30°C for 3–4 days. Germinated seedlings were transplanted into 10 cm pots containing moistened Lambert all-purpose mix (LM-111; Riviere-Ouelle, Quebec, Canada) and placed in a growth chamber with a 14:10 day:night cycle, with a day temperature of 27°C and a night temperature of 24°C. Plants were grown for approximately 5 weeks, fertilized once at first dry down and again approximately 10 days later with 20:20:20 N:P:K (Jack's all-purpose fertilizer, JR Peters, Allentown, PA, USA), and watered as needed.

## Caterpillar sourcing and rearing

Monarch (*D. plexippus*) eggs were obtained from a laboratory colony in late January 2024, and were kept in the lab until hatching. Immediately upon hatching, each caterpillar was placed in a 1 oz deli cup containing one small piece of moist cotton (to prevent leaf tissue from drying) and a leaf disc (0.95 cm diameter) of *A. incarnata* or *A. curassavica*, prepared with the appropriate treatment. Twenty caterpillars were reared on each of nine treatments, consisting of eight treatments applied to *A. incarnata* leaf discs (negative control, five single compound treatments, and two compound mixture treatments) and one treatment applied to *A. curassavica* leaf discs (positive control), for a total of 180 caterpillars. Caterpillars were checked regularly and were given a fresh leaf disc corresponding to their assigned treatment each day or as needed to ensure they did not run out of food. Each time the caterpillar was given a new leaf disc, the previous disc was saved for leaf area determination. On the sixth day of the experiment, we increased the leaf punch diameter size to 1.27 cm, as the caterpillars' feeding increased. All caterpillars were reared in a randomized array on a lab bench, away from a window, at ambient lab temperature.

## Cardenolide solution preparations

Cardenolides from *A. curassavica* foliage were available by previous isolation from aerial material (see methods in *Agrawal et al., 2021*). All compounds were >90% absolute purity by [1]H NMR (except 15-hydroxy-calotropin which was 65% pure) and all compounds were >90% relative purity by UV data at 218 nm. Cardenolides were diluted to approximately 0.5 mM based on estimated amounts: frugoside, calactin, and 15-hydroxy-calotropin in methanol, whereas uscharin and voruscharin were diluted in acetonitrile due to solubility constraints. Each compound was then quantified using an external HPLC-DAD digitoxin calibration curve (in the appropriate solvent) at 218 nm. Both the calibration curve and compound quantification were performed on an Agilent 1100 HPLC (Santa Clara, CA, USA), using the column, specifications, and gradient described below in the 'Cardenolide extractions and HPLC-DAD analysis' section (and also in *Petschenka et al., 2022*). Molar concentrations were converted to mass, based on the known molecular weight of each compound. For single compound treatments, the volume of solution representing 1.2636 mg of each compound was then pipetted into a separate tube and taken to dryness in a rotary evaporator (Labconco, Kansas City, MO, USA). For the equimolar mixture, this same mass (1.2636 mg) was split, in equimolar fashion, between all five compounds, while the realistic mixture contained relative proportions of the five compounds similar to what is found in *A. curassavica* leaf tissue (see *Table 1*). For our realistic measure, we combined knowledge from published studies (*Agrawal et al., 2021*) and unpublished work on the relative abundance of cardenolides in *A. curassavica*. In both cases, the total concentration of the mixtures was equivalent to that of individual compounds. We made a second set of these preparations halfway through the experiment. We acknowledge that the proportions we used (*Table 1*) are based on our experience growing *A. curassavica*, and that the ratios and amounts of these compounds are subject to ontogenetic and environmental influences.

For each treatment, dried cardenolide was brought up in 95% ethanol to a concentration of 0.675 mg/mL. Due to low solubility of some compounds in ethanol, these mixtures were treated as suspensions, vortexed, and sonicated for homogeneity before each use, and stored at 4°C in between uses. Just before each feeding, experimental leaf discs (0.95 cm diameter) were prepared by taking leaf punches directly from fresh leaves in the growth chamber, and then using a pipet to apply 6 µL of the appropriate suspension evenly to the top surface of each leaf punch. Punches were collected from each plant on 1 day only, to prevent any potential effects of induction. Punches were dried in a fume hood and then a second aliquot of 6 µL was applied, for 12 µL total, to bring the concentration of each experimental cardenolide application to 3 mg/g dry leaf tissue, which approximates the total cardenolide concentration of *A. curassavica* leaves based on our previous work and the literature. *A. incarnata* and *A. curassavica* control leaf discs received 12 µL of 95% ethanol alone, in the same manner as above. *A. incarnata* leaves were chosen as the substrate because they are low in cardenolides (mean of 0.0028 mg/g dry mass based on five samples used in this study), while *A. curassavica* leaf discs for this experiment averaged 3.8 mg/g dry leaf tissue. From day 6 on, when larger punches were used, the amount of cardenolide mixture applied to each punch was adjusted to keep the overall cardenolide concentration at 3 mg/g dry leaf tissue.

## Data collection

On each day that leaf discs were changed, notes were taken on caterpillar mortality or molting, and old leaf discs were taped to a piece of paper for estimation of remaining leaf area, using LeafByte (*Getman-Pickering et al., 2020*). Remaining leaf area was subtracted from known starting punch area to estimate total leaf area per day, and these amounts were summed for the 10 days of the experiment. On day 10, the final day of the experiment, caterpillar instar (2 or 3) was recorded, and each caterpillar was weighed and stored at –80°C for subsequent cardenolide analysis. All loose frass was collected from each caterpillar (not counting early instar frass stuck to the cotton piece) and stored at –80°C as well. While we started with 180 caterpillars, 21 caterpillars died during the experiment (final N=141). Caterpillars and frass were freeze-dried prior to extraction.

## Cardenolide extraction and HPLC-DAD analysis

Freeze-dried caterpillar and frass samples were weighed and ground to a powder in a Mixer Mill (Retsch, Haan, Germany) in screw cap tubes (Sarstedt, Nümbrecht, Germany) using 1 stainless steel bead per sample. Samples were extracted as per *Petschenka et al., 2022*, using 1 mL methanol

spiked with 20 µg hydrocortisone standard per sample, and extracted using zirconia-silica beads and a FastPrep-24 (MP Biomedicals, Santa Ana, CA, USA) twice for 45 s at 6.5 m/s. Samples were centrifuged at 20,817×$g$ for 12 min, and each supernatant was transferred to a clean 2 mL tube (caterpillars) or a 1.4 mL racked tube (frass). Caterpillar samples were dried and then brought up in 250 µL methanol, and defatted using 750 µL hexanes, and then taken to dryness, along with the frass samples, in a rotary evaporator (Labconco, Kansas City, MO) at 35°C. Residues were reconstituted in 200 µL methanol, filtered through a hydrophobic filter plate (Millipore, Burlington, MA, USA), and sealed for chromatography analysis.

Fifteen microliters of extract were injected into an Agilent 1100 series HPLC-DAD, and compounds were separated on a Gemini C18 reversed phase column (3 µm, 150×4.6 mm, Phenomenex, Torrance, CA, USA). Cardenolides were eluted on a constant flow of 0.7 mL/min with an acetonitrile-water gradient as follows: 0–2 min 16% acetonitrile, 25 min 70% acetonitrile; 30–40 min 95% acetonitrile, followed by a 10 min post-run in 16% ACN. Peaks were recorded by diode array at 210, 218, 280, 320, and 360 nm, with hydrocortisone as the standard. Using the 218 nm detection data, peaks with symmetrical absorption maxima between 217 and 222 nm were recorded as cardenolides (*Petschenka et al., 2022*). Concentrations were calculated using peak areas of hydrocortisone (and a conversion factor to expected peak area of digitoxin), and total cardenolide concentration was calculated as the sum of all individual cardenolide peaks.

## Statistical approach

To understand the impacts of our individual cardenolide treatments on growth of the caterpillars, we took two approaches. First, on a dry mass basis, we analyzed caterpillar mass as a measure of growth rate, leaf mass ingested, and mass of frass excreted. The effect of cardenolide treatments on the efficiency of conversion of digested matter (EDC) (*Waldbauer, 1968*) was calculated as caterpillar growth/(ingested leaves – frass). One-way ANOVA was used to test the differences between the five cardenolide treatments. In addition, we built models with cardenolide treatment, ingested matter, and their interaction as predictors. These analyses address the extent that feeding impacts growth and digestion differentially among treatments. A parallel set of analyses was conducted on cardenolide sequestration and excretion. Specific contrasts assessed differences between cardenolide types.

To test for a phytochemical diversity effect, our single cardenolide treatments were contrasted to our mixture treatments. Here, we employed fixed effect nested ANOVAs with the five individual cardenolide treatments nested within a 'single' effect and our two mixtures nested within a 'mixture' effect. In this way, the response variable was tested as a function of treatment group (single vs. mixed), with the specific treatments nested within each of these two groups. This approach is functionally equivalent to a contrast of the five single vs. two mixture treatments and also assesses variation among the nested effects. All analyses were conducted using JMP Pro, version 16.

## Estimation of IC-50 of 15-hydroxy-calotropin

The compound identity was verified by NMR (*Supplementary file 1*). Inhibition of both a cardenolide-adapted (monarch butterfly) and an unadapted (porcine) Na/K-ATPase by 15-hydroxy-calotropin was tested using an in vitro enzyme assay described in *Petschenka et al., 2022*. Monarch nervous tissues were dissected, homogenized in Millipore water, and freeze-dried, while the porcine ATPase was obtained commercially (Millipore Sigma, Burlington, MA, USA). A stock solution of our 1 mM 15-hydroxy-calotropin preparation was prepared by HPLC quantification (as described above in the 'Cardenolide solution preparation') in 20% DMSO in Millipore water. Serial dilutions were prepared with 20% DMSO – one at 0.5 mM, then 4 additional 1/10 dilutions, to prepare a 6-point inhibition curve, and each curve was run in triplicate for each enzyme (monarch and porcine), alongside ouabain. Assays were run, as described in *Rubiano-Buitrago et al., 2026*, but using 0.75 monarch brain tissue per mL Millipore water instead of *Oncopeltus* nervous tissue. Absorbances were determined spectrophotometrically at 700 nm, corrected by background values, and dose-response curves were fit using a nonlinear mixed effects model with a four-parameter logistic function in R version 4.5.1, in order to determine the concentration of 15-hydroxy-calotropin that inhibits the activity of each ATPase by 50% (*Rubiano-Buitrago et al., 2026*; *Petschenka et al., 2022*).

## Acknowledgements

We thank Ron White for isolating the cardenolides, Ann Ryan for monarch eggs, Elinor Behlman for technical help in the lab, Christophe Duplais for discussion, and Andrew Siefert from the Cornell Statistical Consulting Unit for advice about the ANOVAs. We thank our lab group (www.herbivory.com) for comments on the project. This research was supported by a grant from the US NSF (IOS-2209762), Federal Capacity Funds allocated to the Cornell Agricultural Experiment Station from the National Institute of Food and Agriculture, and Cornell University.

## Additional information

### Funding

| Funder | Grant reference number | Author |
| --- | --- | --- |
| National Science Foundation | IOS-2209762 | Anurag A Agrawal<br>Amy P Hastings<br>Paola Rubiano-Buitrago |

The funders had no role in study design, data collection and interpretation, or the decision to submit the work for publication.

### Author contributions

Anurag A Agrawal, Conceptualization, Data curation, Formal analysis, Supervision, Funding acquisition, Investigation, Methodology, Writing – original draft, Project administration, Writing – review and editing; Amy P Hastings, Conceptualization, Data curation, Validation, Investigation, Methodology, Writing – review and editing; Paola Rubiano-Buitrago, Formal analysis, Validation, Methodology, Writing – review and editing

### Author ORCIDs

Anurag A Agrawal ![ORCID] https://orcid.org/0000-0003-0095-1220

Reviewer #1 (Public review): https://doi.org/10.7554/eLife.109003.3.sa1
Reviewer #2 (Public review): https://doi.org/10.7554/eLife.109003.3.sa2
Author response https://doi.org/10.7554/eLife.109003.3.sa3

## Additional files

### Supplementary files

Supplementary file 1. 1D NMR assignment of key positions for the identification of 15β-hydroxy-calotropin. Chemical shifts with an asterisk correspond to values in CDCl₃, whereas non-designated ones correspond to values in CD₃OD. Data was compared with published records (*El-Askary et al., 1993*; *Rubiano-Buitrago et al., 2022*).

MDAR checklist

### Data availability

All data have been deposited at https://doi.org/10.7298/sq5m-xx07 and are publicly available.

The following dataset was generated:

| Author(s) | Year | Dataset title | Dataset URL | Database and Identifier |
| --- | --- | --- | --- | --- |
| Agrawal A, Rubiano-Buitrago P, Hastings A | 2025 | Data from: Cardenolide toxin diversity impacts monarch butterfly growth and sequestration | https://doi.org/10.7298/sq5m-xx07 | Cornell University Library eCommons Repository, 10.7298/sq5m-xx07 |

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
