## [Editor Report · eLife Assessment]

This **important** study investigates how structurally diverse cardenolide toxins in tropical milkweed, especially mixtures containing nitrogen- and sulfur-containing variants, influence monarch caterpillar feeding, growth, and toxin sequestration. The experiments provide **solid** evidence that chemical diversity within a single group of plant toxins can have combined effects on even highly specialized herbivores that differ from the effects of each toxin alone. However, as the mixture design does not fully separate true diversity effects from the influence of the N,S-cardenolides themselves and the ecological basis for the chosen natural ratios remains weakly justified. As a result, the broader conclusions would require more fully justified concentration regimes, mixture treatments that exclude N,S-cardenolides, and tests on living plants and non-adapted herbivores to firmly support the proposed coevolutionary interpretation.

---

## [Referee Report · Reviewer #1 (Public review)]

Summary:

In the ecological interactions between wild plants and specialized herbivorous insects, structural innovation-based diversification of secondary metabolites often occurs. In this study, Agrawal et al. utilized two milkweed species (Asclepias curassavica and Asclepias incarnata) and the specialist Monarch butterfly (Danaus plexippus) as a model system to investigate the effects of two N,S-cardenolides-formed through structural diversification and innovation in A. curassavica-on the growth, feeding, and chemical sequestration of D. plexippus, compared to other conventional cardenolides. Additionally, the study examined how cardenolide diversification resulting from the formation of N,S-cardenolides influences the growth and sequestration of D. plexippus. On this basis, the research elucidates the ecophysiological impact of toxin diversity in wild plants on the detoxification and transport mechanisms of highly adapted herbivores.

Strengths:

The study is characterized by the use of milkweed plants and the specialist Monarch butterfly, which represent a well-established model in chemical ecology research. On one hand, these two organisms have undergone extensive co-evolutionary interactions; on the other hand, the butterfly has developed a remarkable capacity for toxin sequestration. The authors, building upon their substantial prior research in this field and earlier observations of structural evolutionary innovation in cardenolides in A. curassavica, proposed two novel ecological hypotheses. While experimentally validating these hypotheses, they introduced the intriguing concept of a "non-additive diversity effect" of trace plant secondary metabolites when mixed-contrasting with traditional synergistic perspectives-in their impact on herbivores.

Weaknesses:

The manuscript has two main weaknesses. First, as a study reliant on the control of compound concentrations, the authors did not provide sufficient or persuasive justification for their selection of the natural proportions (and concentrations) of cardenolides. The ratios of these compounds likely vary significantly across different environmental conditions, developmental stages, pre- and post-herbivory, and different plant tissues. The ecological relevance of the "natural proportions" emphasized by the authors remains questionable. Furthermore, the same compound may even exert different effects on herbivorous insects at different concentrations. The authors should address this issue in detail within the Introduction, Methods, or Discussion sections.

Second, the study was conducted using leaf discs in an in vitro setting, which may not accurately reflect the responses of Monarch butterflies on living plants. This limitation undermines the foundation for the novel ecological theory proposed by the authors. If the observed phenomena could be validated using specifically engineered plant lines-such as those created through gene editing, knockdown, or overexpression of key enzymes involved in the synthesis of specific N,S-cardenolides-the findings would be substantially more compelling.

---

## [Referee Report · Reviewer #2 (Public review)]

I have reviewed both the original and revised version of this manuscript and while no additional experiments were added, I find the interpretations and discussion of the limitations of the study have improved. This is appreciated.

My original concern regarding the mixture treatments largely remains. Figure 4 nicely shows that the mixtures are more potent than the average of all single compounds. However, Fig S3 shows that the effects of mixtures are not significantly different from effects of at least one, single N,S compound (voruscharin or uscharin) across all measured growth/sequestration responses. Essentially, the effects of single N,S compounds is similar to mixtures (which also contain N,S compounds).

While the current results are certainly interesting as presented, in my view the main takeaway of the manuscript would be more compelling if it could be demonstrated that it isn't simply the presence of N,S compounds in the mixtures driving the observations. For example, does a mixture of all compounds except voruscharin or uscharin still have stronger growth/sequestration effects compared to single non-N,S compounds?

---

## [Author Response]

The following is the authors’ response to the original reviews.

**Public Reviews:**

**Reviewer #1 (Public review):**
Summary:In the ecological interactions between wild plants and specialized herbivorous insects, structural innovation-based diversification of secondary metabolites often occurs. In this study, Agrawal et al. utilized two milkweed species (Asclepias curassavica and Asclepias incarnata) and the specialist Monarch butterfly (Danaus plexippus) as a model system to investigate the effects of two N,S-cardenolides - formed through structural diversification and innovation in A. curassavica-on the growth, feeding, and chemical sequestration of D. plexippus, compared to other conventional cardenolides. Additionally, the study examined how cardenolide diversification resulting from the formation of N,S-cardenolides influences the growth and sequestration of D. plexippus. On this basis, the research elucidates the ecophysiological impact of toxin diversity in wild plants on the detoxification and transport mechanisms of highly adapted herbivores.Strengths:The study is characterized by the use of milkweed plants and the specialist Monarch butterfly, which represent a well-established model in chemical ecology research. On one hand, these two organisms have undergone extensive co-evolutionary interactions; on the other hand, the butterfly has developed a remarkable capacity for toxin sequestration. The authors, building upon their substantial prior research in this field and earlier observations of structural evolutionary innovation in cardenolides in A. curassavica, proposed two novel ecological hypotheses. While experimentally validating these hypotheses, they introduced the intriguing concept of a "non-additive diversity effect" of trace plant secondary metabolites when mixed, contrasting with traditional synergistic perspectives, in their impact on herbivores.Weaknesses:The manuscript has two main weaknesses. First, as a study reliant on the control of compound concentrations, the authors did not provide sufficient or persuasive justification for their selection of the natural proportions (and concentrations) of cardenolides. The ratios of these compounds likely vary significantly across different environmental conditions, developmental stages, pre- and post-herbivory, and different plant tissues. The ecological relevance of the "natural proportions" emphasized by the authors remains questionable. Furthermore, the same compound may even exert different effects on herbivorous insects at different concentrations. The authors should address this issue in detail within the Introduction, Methods, or Discussion sections.Second, the study was conducted using leaf discs in an in vitro setting, which may not accurately reflect the responses of Monarch butterflies on living plants. This limitation undermines the foundation for the novel ecological theory proposed by the authors. If the observed phenomena could be validated using specifically engineered plant lines-such as those created through gene editing, knockdown, or overexpression of key enzymes involved in the synthesis of specific N,S-cardenolides - the findings would be substantially more compelling.
**Reviewer #2 (Public review):**
This study examined the effects of several cardenolides, including N,S-ring containing variants, on sequestration and performance metrics in monarch larvae. The authors confirm that some cardenolides, which are toxic to non-adapted herbivores, are sequestered by monarchs and enhance performance. Interestingly, N,S-ring-containing cardenolides did not have the same effects and were poorly sequestered, with minimal recovery in frass, suggesting an alternate detoxification or metabolic strategy. These N,S-containing compounds are also known to be less potent defences against non-adapted herbivores. The authors further report that mixtures of cardenolides reduce herbivore performance and sequestration compared to single compounds, highlighting the important role of phytochemical diversity in shaping plant-herbivore interactions.Overall, this study is clearly written, well-conducted and has the potential to make a valuable contribution to the field. However, I have one major concern regarding the interpretations of the mixture results. From what I understand of the methods, all tested mixtures contain all five compounds. As such, it is not possible to determine whether reduced performance and sequestration result from the complete mixture or from the presence of a single compound, such as voruscharin for performance and uscharin for sequestration. For instance, if all compounds except voruscharin (or uscharin) were combined, would the same pattern emerge? I suspect not, since the effects of the individual N,S-containing compounds alone are generally similar to those of the full mixture (Figure S3). By taking the average of all single compounds, the individual effects of the N,S-containing ones are being inflated by the non-N,S-containing ones (in the main text, Figure 4). In the mix, of course, they are not being 'diluted', as they are always present. This interpretation is further supported by the fact that in the equimolar mix, the relative proportion of voruscharin decreases (from 50% in the 'real mix'), and the target measurements of performance and sequestration tend to increase in the equimolar mix compared to the real mix.Despite this issue, the discussion of mixtures in the context of plant defence against both adapted and non-adapted herbivores is fascinating and convincing. The rationale that mixtures may serve as a chemical tool-kit that targets different sets of herbivores is compelling. The non-N,S cardenolides are effective against non-adapted herbivores and the N,S-containing cardenolides are effective against adapted herbivores. However, the current experiments focus exclusively on an adapted species. It would be especially interesting to test whether such mixtures reduce overall herbivory when both adapted and non-adapted species are present.It remains possible that mixtures, even in the absence of voruscharin or uscharin, genuinely reduce sequestration or performance; however, this would need to be tested directly to address the abovementioned concern.

Thanks for these insightful reviews and your summary assessment. We certainly agree that ours was a laboratory study with a single specialized insect, and both mixtures types had all five compounds (controlling for total toxin concentration). Thus, our conclusion that combined effects of naturally occurring toxins (within the cardenolide class) have non-additive effects for the specialized sequestering monarch are constrained by our experimental conditions. In our assay we used two mixture types, equimolar and “natural” proportions. We acknowledge that the natural proportions will vary with plant age, damage history, etc. of the host plant, Asclepias curassavica. Our proportions were based on growing the plants a few different times under variable conditions. Although we did not conduct these experiments on non-adapted insects, we discuss a related experiment that was conducted with wild-type and genetically engineered Drosophila (Lopez-Goldar et al. 2024, PNAS). In sum, we appreciate the reviewers’ comments.

**Recommendations for the authors:**

**Reviewing Editor Comments:**
(i) More convincingly justify the choice and ecological relevance of the "natural" cardenolide ratios, (ii) Clarify the interpretation of mixture effects, and (iii) more explicitly discuss the limitations of leaf-disc assays and the absence of non-adapted herbivores in light of the broader coevolutionary claims.

Thank you for these suggestions. We have added several sentences of text to the Discussion section to make these points.

**Reviewer #1 (Recommendations for the authors):**
(1) Statistical analysis is missing from Figure 3 and Figure S3, making it difficult to assess the significance of the data.

Much of the data in Fig. 3 is meant for descriptive presentation, with the main statistical analysis contrast between N,S and non-N,S cardenolides given in the main text of the results. We have added treatment differences between the sequestration efficiencies to the figure as well.

(2) To help readers intuitively understand how certain results (such as ECD and sequestration efficiency) were calculated, the authors can provide the equations used for these computations.

Thank you, this was given in the methods and we have added it to the Result on first mention as well.

(3) For Figure 4, we suggest presenting the results of the equal mixture treatment and the realistic mixture treatment separately, rather than averaging the results from these two types of treatments.

We understand and appreciate this comment – all of the treatment means are given in Fig. S3. For this particular figure we have opted to stick with the binary comparison (singles vs. mixed) to maximize replication for statistical tests (typically n = 25 vs. 10).

**Reviewer #2 (Recommendations for the authors):**
Given the interpretations and discussion generally, I feel the manuscript would benefit from either additional experiments (mixtures w/o N-S compounds), inclusion of non-adapted herbivore performance, or reframing of the explicit interpretations from your findings.

We have added some caveats to the text but not added any additional experiments.

Also, for all treatments/mixtures are concentrations above the IC50? Perhaps this could be calculated from the information presented, but it may be best to explicitly mention this.

This is an interesting question. IC50’s are estimated from in vitro assays (with the enzyme and toxins in microplate wells) and so are not translatable to foliar concentrations. As indicated in the text, we chose cardenolide levels based on foliar concentrations to match *A. curassavica*.

Some minor points:(1) Although the intact N,S-ring-containing compounds are recovered in low amounts in frass (and not sequestered), is there evidence of N,S-ring components being otherwise traceable in the frass? For example, can excess S or N be detected in frass? This could provide insight into differential detoxification or reincorporation of these elements, potentially explaining variation between voruscharin and uscharin.

Great question! We have not been able to detect breakdown projects. In other experiments we have conducted mass spectrometric analysis of bodies and frass, but have not been able to find the features representing breakdown products. Nonetheless, as mentioned below, the main conversion products are evident and measurable, as in this study.

(2) As a point of curiosity, is there evidence of interconversion between such compounds? For instance, if monarchs are fed only voruscharin, can other cardenolides be detected in their tissues?

Yes, we have tried to make this more clear in the text. Both uscharin and voruscharin are converted to calotropin and calactin.